# Comparison of the Relationship between Visual Acuity and Motor Function in Non-Elderly and Elderly Adults

**DOI:** 10.3390/jcm12052008

**Published:** 2023-03-03

**Authors:** Sadayuki Ito, Hiroaki Nakashima, Masaaki Machino, Naoki Segi, Shinya Ishizuka, Yasuhiko Takegami, Jun Takeuchi, Jun Ouchida, Yukiharu Hasegawa, Shiro Imagama

**Affiliations:** 1Department of Orthopaedic Surgery, Nagoya University Graduate School of Medicine, Nagoya 466-8560, Japan; 2Department of Ophthalmology, Nagoya University Graduate School of Medicine, Nagoya 466-8550, Japan; 3Department of Rehabilitation, Kansai University of Welfare Science, Osaka 582-0026, Japan

**Keywords:** back muscle strength, gait speed, motor function, visual acuity

## Abstract

This study aimed to clarify the relationship between visual acuity and motor function in younger and elderly participants and to compare differences between non-elderly and elderly participants. In total, 295 participants who underwent visual and motor functional examinations were included; participants with visual acuity ≥0.7 were assigned to the normal group (N group) and those with visual acuity ≤0.7 were assigned to the low-visual-acuity group (L group). Motor function was compared between the N and L groups; the analysis was performed by grouping participants into those aged >65 years (elderly) and those aged <65 years (non-elderly). The non-elderly group (average age, 55.6 ± 6.7 years) had 105 and 35 participants in the N and L groups, respectively. Back muscle strength was significantly lower in the L group than in the N group. The elderly group (average age, 71.1 ± 5.1 years) had 102 and 53 participants in the N and L groups, respectively. Gait speed was significantly lower in the L group than in the N group. These results reveal differences in the relationship between vision and motor function in non-elderly and elderly adults and indicate that poor vision is associated with lower back-muscle strength and walking speed in younger and elderly participants, respectively.

## 1. Introduction

Sensory organs play an important role in motor function [1]. Therefore, when considering declines in motor function, it is necessary to evaluate muscle strength and sensory organs. Although motor function is often the focus of attention with regard to fall risk in the elderly, sensory organs also play an important role [2]. Therefore, it is important to clarify the relationship between motor function and sensory organs in terms of motor function decline and falls. Although some studies on the relationship between motor function and sensory organs have been conducted, this relationship remains unclear [1].

Visual function has a significant impact on balance and mobility and is closely related to locomotion and falls [1]. Motor and visual functions decline with age [3,4]. Physical activity also plays an important role in visual and motor functions [5,6]. Thus, common factors affecting motor and visual functions have been observed.

The form of visual impairment differs with age group, with myopia in younger age groups and presbyopia in older age groups [7]. Therefore, the effect of visual acuity on motor function is likely to differ with age. However, no large-scale reports have compared visual acuity and motor function in different age groups. Therefore, if the relationship between visual acuity and motor function in each age group can be clarified, it will be possible to establish effective age-appropriate prevention and treatment methods for both young and elderly individuals. This will make it possible to effectively prevent and treat the decline in vision and motor functions for all ages, not only for those who are at risk of falling at present but also for those who are expected to be at higher risk of falling in the future. This can significantly reduce the number of individuals who will fall in the future.

Visual acuity is relatively stable until the age of 65 years [8], and the form of visual acuity loss differs between those aged <65 years and those aged >65 years. Therefore, a comparison of the relationship between visual acuity and motor function between those aged <65 years and >65 years would help clarify the relationship between visual acuity and motor function with age.

Therefore, this study aimed to clarify the relationship between visual acuity and several motor functions (muscle strength, gait speed, and body balance) with age by examining the relationship between motor function and visual acuity in individuals aged <65 years and >65 years using a large cohort and to compare the differences in the relationship between non-elderly and elderly participants.

## 2. Materials and Methods

### 2.1. Study Participants

The surveyed individuals were volunteers who underwent a municipal-supported health checkup in Yakumo in 2019. Yakumo has a population of approximately 17,000, of whom 28% are aged >65 years. More individuals in Yakumo are engaged in agriculture and fishing than in urban areas. The town has been conducting annual health checkups since 1982. Physical examinations include voluntary orthopedic and physical function tests, internal examinations, psychological examinations, and a health-related quality of life survey (SF-36) [9]. This study included all participants who completed an assessment of visual acuity, bioelectrical impedance analysis (BIA), back muscle strength, gait speed, and stabilometry. The exclusion criteria were a history of spine or joint surgery, severe knee injury, severe hip osteoarthritis, a history of hip or spine fracture, neuropathy, severe mental illness, diabetes, kidney or heart disease, non-fasting, severe impairment of walking or standing, and impairment of the central or peripheral nervous system.

Of the 537 participants who underwent health checkups, 295 (117 men and 178 women) met the inclusion criteria. The research protocol was approved by the Human Research Ethics Committee and the university’s Institutional Review Board (No. 2014-0207). All participants provided written informed consent before participation. The research procedure was conducted in accordance with the principles of the Declaration of Helsinki.

### 2.2. Visual Acuity

The best-corrected visual acuity (BCVA) was measured using a standard Japanese visual acuity chart. Decimal BCVA was converted to the logarithm of the minimum angle of resolution (logMAR) for statistical analyses [10].

The risk of traffic accidents increases when visual acuity is <0.7 [11]. In addition, in Japan, a minimum visual acuity of 0.7 is required to obtain a driver’s license. Therefore, participants with a visual acuity of 0.7 or better were assigned to the normal visual acuity group (N group), and those with a visual acuity of <0.7 were assigned to the low visual acuity group (L group).

### 2.3. Bioelectrical Impedance Analysis (BIA)

Anthropometric data, including height, weight, body mass index (BMI), and skeletal muscle mass index (SMI) of each limb, were measured using BIA. An InBody 770 BIA device (Inbody Co., Ltd., Seoul, Republic of Korea), which can differentiate tissues based on their electrical impedance, was utilized [12]. BMI was calculated using the following formula: weight (kg)/height^2^ (m^2^). SMI was calculated using the following formula: SMI  =  appendicular skeletal muscle mass (kg)/height^2^ (m^2^) [13]. All cases were calculated with the same model.

### 2.4. Examination of Motor Function

#### 2.4.1. Back Muscle Strength

For back muscle strength, the maximum isometric strength of the back extensor muscles was measured in the standing position with lumbar flexion of 30° and knees extended using a digital T.K.K. 5102 dynamometer (Takei Co., Tokyo, Japan). The average force from the two trials was recorded, and the maximum strength in each trial was measured [14]. 

#### 2.4.2. Gait Speed

The 10 m gait time was measured to evaluate mobility as the time required to complete a 10 m straight course at the fastest pace possible for each participant. The participants performed the test twice, both at maximum pace, and the mean time was used for analysis [15]. All cases were performed in the same lane.

#### 2.4.3. Stabilometry

We used a Gravicorder GW-5000 (Anima, Tokyo, Japan) for stabilometric recordings. The participants were instructed to step at the center of the device with their feet together for 60 s with their eyes open. The same procedure was repeated with their eyes closed. The device contained vertical force transducers that were used to measure instantaneous fluctuations in the center-of-pressure (COP) values. Outcome measures included the total length of COP sway and root mean square of the area traced by the COP sway. The measured data were recorded at 100 Hz [16]. We used the length of COP sway per second and the surrounding area of the COP sway. Higher values for both the length of COP sway per second and the surrounding area of the COP sway indicate that the trunk balance is unstable. All cases were performed with the same model.

### 2.5. Statistical Analyses

Continuous variables are expressed as mean ± standard deviation. We compared the continuous variables of the L group with those of the N group using Student’s t-test and categorical variables of the L group with those of the N group using the chi-squared test. Each analysis was separately performed for the non-elderly (<65 years) and elderly (>65 years) groups.

Logistic regression analysis was performed to separately evaluate the important predictors of the L group for the non-elderly and elderly groups. The dependent variable was the N versus the L group. Following univariate analysis (sex, age, BFP, SMI, back muscle strength, gait speed, length of COP sway (eyes open), length of COP sway (eyes closed) surrounding area of COP sway (eyes open), and surrounding area of COP sway (eyes closed)), variables that yielded a *p* value < 0.05, sex, age, and BMI, were included in the multivariable analysis.

All statistical analyses were performed using SPSS Statistics version 29.0 software for Mac (IBM Corp., Armonk, NY, USA). Statistical significance was set at *p* value < 0.05.

## 3. Results

The participant characteristics are shown in Table 1. There were 117 male and 178 female participants, with an average age of 63.7 ± 9.7 years. There were 207 and 88 participants in the N and L groups, respectively. The mean SMI, back muscle strength, and gait speed were 6.81 ± 1.04 kg/m^2^, 81.1 ± 32.7 kg, and 2.06 ± 0.35 m/s, respectively.

### 3.1. Non-Elderly Participants

In the non-elderly group, the average age was 55.6 ± 6.7 years. There were 105 and 35 participants in the N and L groups, respectively. The mean SMI, back muscle strength, and gait speed were 6.79 ± 1.04 kg/m^2^, 81.0 ± 33.8 kg, and 2.14 ± 0.34 m/s, respectively (Table 1).

Back muscle strength was significantly lower in the L group than in the N group (N: 84.4 ± 34.5, L: 70.8 ± 30.1, *p* = 0.043). Sex, age, BMI, gait speed, length of COP sway (eyes open and closed), and surrounding area of COP sway (eyes open and closed) were not significantly different between the groups (Table 2).

Next, muscle strength, sex, age, and BMI were examined as covariates for predictors for the L group in logistic regression analysis, which showed sex and back muscle strength to be risk factors for the L group (sex: Exp(B) 0.121, 95% confidence interval (CI): 0.022–0.663, *p* = 0.015; back muscle strength: Exp(B) 0.960, 95% CI: 0.934–0.986, *p* = 0.003) (Table 3).

Because of the significant differences by sex, univariate analysis of back muscle strength by sex was performed, and back muscle strength was significantly lower in the L group than in the N group for both sexes (male: N: 124.1 ± 24.7, L: 105.5 ± 22.6, *p* = 0.042; female: N: 63.9 ± 15.6, L: 53.4 ± 13.5, *p* = 0.007) (Appendix A).

### 3.2. Elderly Participants

In the elderly group, the average age was 71.1 ± 5.1 years. There were 102 and 53 participants in the N and L groups, respectively. The mean SMI, back muscle strength, and gait speed were 6.82 ± 1.05 kg/m^2^, 81.3 ± 31.9 kg, and 2.00 ± 0.34 m/s, respectively (Table 1).

The average age was significantly higher in the L group than in the N group (N: 70.4 ± 4.2, L: 72.4 ± 6.2, *p* = 0.037). Gait speed was significantly lower in the L group than in the N group (N: 2.06 ± 0.32, L: 1.87 ± 0.36, *p* = 0.002). Sex, BMI, back muscle strength, length of COP sway (eyes open and closed), and surrounding area of COP sway (eyes open and closed) were not significantly different between the groups (Table 4).

Next, gait speed, age, sex, and BMI were examined as covariates for risk factors for the L group in the logistic regression analysis, which showed only gait speed to be a risk factor for the L group (Exp(B) 0.136, 95% CI: 0.035–0.525, *p* = 0.004) (Table 5).

In addition, univariate analysis was separately performed for elderly men and elderly women. Gait speed was significantly lower in the L group than in the N group for both sexes (male: N: 2.1 ± 0.3, L: 2 ± 0.4, *p* = 0.024; female: N: 1.9 ± 0.2, L: 1.7 ± 0.2, *p* = 0.009). In elderly men, age was significantly higher in the L group than in the N group (N: 70.8 ± 3.9, L: 73.8 ± 5.8 *p* = 0.012) (Appendix A).

## 4. Discussion

There have been several reports on vision and motor function. Some studies on the elderly have reported that visual acuity is associated with grip strength, muscle mass, and falls [1,6]. Moreover, a study in young adults reported an association between myopia and physical activity [17]. However, no studies have reported the relationship between visual acuity and motor function according to age group. This is the first study to examine the relationship between visual acuity and motor function separately for participants aged <65 years and >65 years and to compare the differences in the relationship between non-elderly and elderly participants. The results of this study revealed an association between visual acuity and back muscle strength in participants aged <65 years and an association between visual acuity and walking speed in participants aged >65 years, indicating differences in age-related and non-age-related associations between motor function and visual impairment.

Various risk factors for myopia have been reported in young people [17,18]. A short time spent outside and short exposure to sunlight during school age are risk factors for myopia [17,18]. One mechanism by which bright light prevents myopia is that light stimulates the release of dopamine in the retina, and this neurotransmitter inhibits the stretching of the developing eye. This was demonstrated by showing that the myopia-preventive effect of bright light disappears when a dopamine-suppressing drug, spiperone, is injected into the eyes of chickens [19]. As for the muscle, it is related to sunlight in terms of vitamin D metabolism [20]. Moreover, time spent in the sun is related to outdoor activity and may be associated with increased activity [21]. Both myopia and muscle in young adults may be associated with sunlight. These reports may explain why the results of the present study found an association between vision loss and back muscle weakness in the non-elderly group.

Regarding back muscle strength, a study [22] performed by occupation demonstrated that it was significantly lower in video display terminal (VDT) operators who were performing near work. VDT operators work in a static sitting posture, which decreases blood flow, depletes nutrients, and may cause metabolic waste products to accumulate in the back. Moreover, static work causes muscle fatigue faster than dynamic work [22]. As such, VDT operators have low back muscle strength scores [22]. Thus, myopia and back muscle strength in young adults are influenced by near work and the duration of that work, which may be closely related. In the present study, the association between vision loss and back muscle weakness was also observed in the non-elderly group, and it is possible that near work and its duration may be two of the factors contributing to the results of this study. Therefore, adjusting work hours and the work environment may help prevent myopia and back muscle weakness in young people.

The incidence of visual impairment increases with age [4]. The age-related visual impairment, occurring noticeably at around age 40 and then at around age 60, is attributed to neuronal cell loss and increased neural variability, most likely higher than the primary visual cortex [23]. Sensory deficits are associated with several problems that occur during old age. Studies have mainly reported an association between vision and an increased frequency of falls and an increased risk of recurrent falls in elderly women due to visual impairment [24]. Moreover, greater frailty is associated with poor visual function [25]. Visual acuity plays a leading role in balance control by providing the nervous system with continuously updated information about the position and movement of body segments in relation to each other and the environment [26]. Therefore, it may influence gait speed. The results of the present study, which found the association between walking speed and visual acuity in the elderly, were consistent with these reports.

Moreover, with regard to the association between visual acuity and walking speed in the elderly observed in the present study, elderly adults with reduced visual acuity tend to avoid physical activity for fear of falling, and avoidance of physical activity causes muscle weakness. Therefore, prevention of reduced visual acuity may reduce the decline in activity and prevent the decline in motor function. Perceptual learning methods are effective in improving visual performance in the elderly [27], and such visual improvement training may be effective in elderly people with significantly reduced motor function and a high risk of falling.

In addition, the association between visual acuity and walking speed in the elderly observed in the results of the this may be related to age-related visual impairment and motor dysfunction, which may share more common causes than aging, vascular inflammation, or disease. In experiments with mice, exercise reduced abnormal choroidal blood vessels, suggesting that exercise may inhibit vascular inflammation, possibly inhibiting the progression of age-related visual impairment [28]. Therefore, exercise may help prevent visual impairment and improve gait speed.

In non-elderly participants, visual acuity loss is considered simple myopia, and no loss of other visual functions is considered to have occurred. Walking speed has been associated with normal visual acuity and contrast visual acuity [29]. Therefore, it is likely that visual acuity is not associated with walking speed in younger participants, and there may have been no association between visual acuity and walking speed in non-elderly participants in this study.

It is likely that vision loss in the elderly is not simply myopia but is accompanied by several age-related declines in visual function. Therefore, in older adults, visual acuity may not be associated with back muscle strength, which is associated with myopia. Thus, vision loss in younger patients may be simple myopia, whereas vision loss in elderly patients may be accompanied by various visual function deficits, which may have been the cause of the motor function differences found in this study that were associated with vision loss in the non-elderly and elderly.

This study has some limitations. First, we were not able to examine visual functions other than visual acuity. Second, we were not able to assess activity because we were unable to ask about lifestyle habits. Third, the finding of higher back strength in the elderly than in the non-elderly is contrary to basic knowledge. In sex-specific analysis, muscle strength was higher in the non-elderly group than in the elderly group, suggesting that gender differences may have influenced the results. However, in the male group with low visual acuity, the average muscle strength was almost the same for non-elderly and elderly group. In the female group with low visual acuity, mean muscle strength was higher in the elderly group than in the non-elderly group. The reason for this may be that myopia, which is associated with reduced back muscle strength due to VDT work and other environmental factors, was more common in the non-elderly group, while presbyopia, which is less associated with such environmental factors, may have been a cause of low visual acuity in the non-elderly group. It should be noted that the participant population was from one region and may have been a special population. Fourth, in this study, we did not have enough information on occupational history and lifestyle, and we were not able to examine whether there was vision loss or differences in occupational history and lifestyle between younger and older adults. Finally, a longitudinal study should be conducted to compare changes between younger and elderly adults despite the present study utilizing a cross-sectional design.

## 5. Conclusions

The results of this study reveal differences in the relationship between visual acuity and motor function in non-elderly and elderly individuals. In non-elderly participants, visual acuity may be associated with back muscle strength, and in elderly participants, visual acuity may be associated with gait speed. This implies that biological or longitudinal prospective studies are needed to understand the effect of decline of sensory organ function on reduced muscle strength. Moreover, in older people clinically diagnosed with sensory function decline, it may be necessary to develop a program to prevent future muscle weakness.

## Figures and Tables

**Table 1 jcm-12-02008-t001:** Comparison of each parameter between non-elderly and elderly participants.

	Total (*n* = 295)	Non-Elderly (*n* = 140)	Elderly (*n* = 155)
Visual acuity (N/L)	207/88	105/35	102/53
Male/female	117/178	45/95	72/83
Age (yrs)	63.7 ± 9.7	55.6 ± 6.7	71.1 ± 5.1
BMI (kg/m^2^)	23.7 ± 3.6	23.8 ± 3.8	23.6 ± 3.4
BFP (%)	29.2 ± 7.6	30.5 ± 7.6	28.0 ± 7.4
SMI (kg/m^2^)	6.81 ± 1.04	6.79 ± 1.04	6.82 ± 1.05
Back muscle strength (kg)	81.1 ± 32.7	81.0 ± 33.8	81.3 ± 31.9
Gait speed (m/s)	2.06 ± 0.35	2.14 ± 0.34	2.00 ± 0.34
Length of COP sway (eyes open) (cm/s)	1.63 ± 0.55	1.47 ± 0.42	1.77 ± 0.61
Length of COP sway (eyes closed) (cm/s)	2.04 ± 0.93	1.86 ± 0.8	2.22 ± 1.01
Surrounding area of COP sway (eyes open) (cm^2^)	2.64 ± 1.51	2.50 ± 1.38	2.77 ± 1.61
Surrounding area of COP sway (eyes closed) (cm^2^)	3.22 ± 2.43	2.99 ± 2.27	3.44 ± 2.56

Visual acuity (N/L), normal visual acuity group (N group)/low visual acuity group (L group); BMI, body mass index; BFP, body fat percentage; SMI, skeletal muscle mass index, COP, center of pressure.

**Table 2 jcm-12-02008-t002:** Comparison of each parameter between the N and L groups in non-elderly participants.

Non-Elderly	N (*n* = 105)	L (*n* = 35)	*p* Value
Male/female	35/70	10/25	0.679
Age (yrs)	55.4 ± 6.8	56.4 ± 6.4	0.424
BMI (kg/m^2^)	23.6 ± 3.7	24.3 ± 4.0	0.381
BFP (%)	30.1 ± 7.7	31.9 ± 7.5	0.214
SMI (kg/m^2^)	6.79 ± 1.04	6.79 ± 1.03	0.996
Back muscle strength (kg)	84.4 ± 34.5	70.8 ± 30.1	0.043 *
Gait speed (m/s)	2.17 ± 0.36	2.05 ± 0.26	0.05
Length of COP sway (eyes open) (cm/s)	1.45 ± 0.43	1.54 ± 0.40	0.294
Length of COP sway (eyes closed) (cm/s)	1.84 ± 0.79	1.90 ± 0.83	0.712
Surrounding area of COP sway (eyes open) (cm^2^)	2.37 ± 1.36	2.88 ± 1.37	0.068
Surrounding area of COP sway (eyes closed) (cm^2^)	2.92 ± 2.27	3.18 ± 2.29	0.572

Visual acuity(N/L), normal visual acuity group (N group)/low visual acuity group (L group); BMI, body mass index; BFP, body fat percentage; SMI, skeletal muscle mass Index, COP, center of pressure. * *p* < 0.05.

**Table 3 jcm-12-02008-t003:** Logistic regression analysis for risk factors of low visual acuity (L group) in non-elderly participants.

Non-Elderly	B	SE	Wald	df	*p*	Exp (B)	95% CI
Sex	−2.11	0.867	5.921	1	0.015 *	0.121	0.022–0.663
Age	0.017	0.037	0.208	1	0.648	1.017	0.946–1.094
BMI	0.044	0.061	0.524	1	0.469	1.045	0.927–1.178
Back muscle strength	−0.041	0.014	9.081	1	0.003 *	0.960	0.934–0.986

BMI, body mass index. * *p* < 0.05.

**Table 4 jcm-12-02008-t004:** Comparison of each parameter between the N and L groups in elderly participants.

Elderly	N (*n* = 102)	L (*n* = 53)	*p* Value
Male/female	48/54	24/29	0.866
Age (yrs)	70.4 ± 4.2	72.4 ± 6.2	0.037 *
BMI (kg/m^2^)	23.2 ± 3.4	24.2 ± 3.5	0.088
BFP (%)	27.4 ± 7.0	29.1 ± 8.0	0.206
SMI (kg/m^2^)	6.83 ± 1.03	6.81 ± 1.10	0.919
Back muscle strength (kg)	81.1 ± 32.0	81.5 ± 32.1	0.947
Gait speed (m/s)	2.06 ± 0.32	1.87 ± 0.36	0.002 *
Length of COP sway (eyes open) (cm/s)	1.76 ± 0.54	1.80 ± 0.72	0.743
Length of COP sway (eyes closed) (cm/s)	2.21 ± 0.94	2.23 ± 1.15	0.901
Surrounding area of COP sway (eyes open) (cm^2^)	2.73 ± 1.51	2.86 ± 1.78	0.653
Surrounding area of COP sway (eyes closed) (cm^2^)	3.24 ± 2.32	3.79 ± 2.93	0.243

Visual acuity(N/L), normal visual acuity group (N group)/low visual acuity group (L group); BMI, body mass index; BFP, body fat percentage; SMI, skeletal muscle mass index, COP, center of pressure. * *p* < 0.05.

**Table 5 jcm-12-02008-t005:** Logistic regression analysis for risk factors of low visual acuity (L group) in elderly participants.

Elderly	B	SE	Wald	df	*p*	Exp (B)	95% CI
Sex	−0.274	0.417	0.431	1	0.511	0.76	0.336–1.722
Age	0.065	0.038	2.865	1	0.091	1.067	0.990–1.150
BMI	0.035	0.057	0.373	1	0.541	1.036	0.925–1.159
Gait speed	−1.995	0.689	8.377	1	0.004 *	0.136	0.035–0.525

BMI, body mass index. * *p* < 0.05.

## Data Availability

The health-checkup data used to support the findings of this study are available from the corresponding author upon request.

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
