# Peer review of "Comparison of the Relationship between Visual Acuity and Motor Function in Non-Elderly and Elderly Adults"

_jcm, 2023, doi:10.3390/jcm12052008_

Round 1
Reviewer 1 Report
This manuscript is about association between motor and visual functions. This may be an interesting and largely unexplored field.
I, however, have some comments.
1. I do not think that Younger and Elderly Adults should be used for these groups. I read Young adults as adults between 20 and 30 yrs. And I have read about elderly workers, being above 55 yrs.
2. The introduction early includes a statement, which I cannot trace clearly in the cited reference [3]. That article tries to explain why so many young people now have myopia. That needs to be clarified or put in another way. (Has by mistake wrong reference been used?)
3. The aim “Therefore, this study aimed to clarify the relationship between visual acuity and motor function with age…” looks too wide to me, considering what is actually studied. I think that “motor function” should be specified
4. Also in the discussion there is an unsatisfactory citation to [3]: “Furthermore, near work is a risk factor for myopia, with near reading distance and continuous working hours being particularly important.[3] Near reading brings about a hyperopic defocus in the eye owing to its close proximity to the page.”
I read in [3]: “Attractive though the idea was, it did not hold up. In the early 2000s, when researchers started to look at specific behaviours, such as books read per week or hours spent reading or using a computer, none seemed to be a major contributor to myopia risk. But another factor did.”
This gives me doubts about how careful the authors overall are.
5. Also in the discussion it reads (line 218):
“Thus, myopia and back muscle strength in young adults are influenced by near work and the duration of that work and may be closely related.”
Then in line 251 it reads: “Therefore, a shorter near reading distance and longer continuous working time may not have caused visual acuity loss and may not have been associated with back muscle strength.”
Together these sentences make a blurry understanding.
6. It is hard also to understand, and maybe also hard for the authors to explain, why the there was no sign of a difference in back muscle strength, between N and L, in the older group. Could that difference, in the younger group, be connected to something else than visual acuity? Maybe more office workers?
7. The younger group (mean age, 56):
|
N |
L |
p |
|
|
Back muscle strength (kg) |
84.4 ± 34.5 |
70.8 ± 30.1 |
0.043* |
The older group (mean age, 71):
|
|
|
||
|
Back muscle strength (kg) |
81.1 ± 32.0 |
81.5 ± 32.1 |
0.947 |
In average, strength declines with age, but here there are small differences between the age groups. And, for L, it is higher in the older group. That could be, I suppose, due to differences in male/female proportions. But maybe, when strength is in focus, they analyses should be done separately per sex?
8. The conclusions from the study are written as “The results of this study reveal differences in the relationship between visual acuity and motor function in younger and elderly individuals. In younger participants, visual acuity is associated with back muscle strength, and in elderly participants, visual acuity is associated with gait speed."
Considering my comment on reasons for low back strength differences, can it be written “is” in “visual acuity is associated with back muscle strength” from the results, which did not include that many possible confounders? I do not think so.
The second part of the sentence is about gait speed. I believe that is a true conclusion, and it has been seen before that people with impaired vision are generally less physical active.
We can compare that second part with the last part of conclusions of [1]:
"This implies that biological or longitudinal prospective studies are needed to understand the effect of decline of sensory organ function on reduced muscle strength. Moreover, in older people clinically diagnosed with sensory function decline, it may be necessary to develop a program to prevent future muscle weakness."
If comparing the conclusions, what can we say is the scientific contribution from this study?
Reviewer 2 Report
This is a comparative and predictive cross-sectional study investigating the relationship between visual acuity and motor function. Authors often use the term “risk factors” for what they should be calling “predictors” or “independent variables”. Also, I find the Discussion somewhat focusing more on results from other studies, which are not clear how they relate to the findings of the current study and should be revised. And careful when writing potential mechanisms or cause-effect assumptions of your findings. Your design does not allow for that nor your outcome measures, hence are overly speculative.
Specific comments:
Title and throughout the manuscript: Perhaps “non-elderly” suits better throughout, i.e., more indicative of your design and sample than “younger adults”.
Line 88: Add immediately after the subheading, the acronym “(BIA)”
Lines 89–106: Provide details such as if the tests are to be replicated (all tests), the units of measurement (all tests), data processing (e.g., stabilometry) and how to interpret the outcome measures (e.g., COP sway)
Line 122: I suggest “predictors” instead of “risk factors”.
Line 124: Which univariate analyses were conducted? Correlation? Which tests?
Line 152: I suggest “predictors” instead of “risk factors”.
Line 189: “large-scale” sounds exaggerated, particularly because you do not present a sample size calculation for conducting this study.
Line 214: what “report” are the authors referring to?
Line 219: This sentence “As such, VDT operators have low back muscle strength scores.” needs a reference.
Line 222: Are these “young people” within the age range of your sample? Please explain how the findings you are reporting from other studies relate with the findings of the current study.
Line 224: “higher than” what?
Line 229: Considering your study measurements/outcome measures and results what is the relevance of this statement? (“Moreover, higher frailty, is associated with poor visual function”)
Round 2
Reviewer 1 Report
Thank you for your improvements! I think that the manuscript is better.
For my comment nr 7, which was:
7. The younger group (mean age, 56):
|
|
N |
L |
p |
|
Back muscle strength (kg) |
84.4 ± 34.5 |
70.8 ± 30.1 |
0.043* |
The older group (mean age, 71):
|
|
|
|
|
|
Back muscle strength (kg) |
81.1 ± 32.0 |
81.5 ± 32.1 |
0.947 |
In average, strength declines with age, but here there are small differences between the age groups. And, for L, it is higher in the older group. That could be, I suppose, due to differences in male/female proportions. But maybe, when strength is in focus, they analyses should be done separately per sex?
You replied: We added results and supplement tables for the elderly as well as the non-elderly.
Thank you!
But the measured strength was higher in the elderly group (for both sexes), which is against basic knowledge that muscle strength declines, not increase, with age. I supposed that the differences could be due to differences in male/female proportions.
So I did suppose that the lower strength, in the non-elderly group, in the overall result was due to a higher fraction of women in the non-elderly group (which would give a low average strength).
However, in your newly added supplement tables, for both men and women, non-elderly subjects are in average weaker than the elderly (see below). This makes the reliability of the measurements, and/or the representativeness of the groups, low.
This should also be mentioned in the limitations paragraph, i.e. that the higher strength in elderly group was surprising and unusual, and therefore the found significant difference in between N and L, should be interpreted carefully.
(I appreciate that you now are more careful in your conclusions.)
|
Non-elderly men |
Total (n = 45) |
N (n = 35) |
L (n = 10) |
P value |
|
Back muscle strength (kg) |
119.5 ± 25.3 |
124.1 ± 24.7 |
105.5 ± 22.6 |
0.042* |
|
elderly men |
total(n=72) |
N(n=48) |
L(n=24) |
P value |
|
Back muscle strength(kg) |
106.7±24 |
106.7±23.6 |
106.6±25.4 |
0.963 |
|
Non-elderly women |
Total (n = 95) |
N (n = 70) |
L (n = 25) |
P value |
|
Back muscle strength (kg) |
61.2 ± 15.7 |
63.9 ± 15.6 |
53.4 ± 13.5 |
0.009* |
|
elderly women |
total(n=83) |
N(n=54) |
L(n=29) |
P value |
|
Back muscle strength(kg) |
58.5±17.6 |
57.5±16.8 |
60.4±19.3 |
0.514 |
Reviewer 2 Report
In general, I´m satisfied with specific amendments authors have performed in the manuscript. However, I maintain my general appraisal to the Discussion section:
– I find the Discussion somewhat focusing more on results from other studies, which are not clear how they relate to the findings of the current study;
– Writing style often leads the reader to assume cause-effect relationship of the findings when the study design does not allow such inference.
